# International External Validation of Risk Prediction Model of 90-Day Mortality after Gastrectomy for Cancer Using Machine Learning

**DOI:** 10.3390/cancers16132463

**Published:** 2024-07-05

**Authors:** Mariagiulia Dal Cero, Joan Gibert, Luis Grande, Marta Gimeno, Javier Osorio, Maria Bencivenga, Uberto Fumagalli Romario, Riccardo Rosati, Paolo Morgagni, Suzanne Gisbertz, Wojciech P. Polkowski, Lucio Lara Santos, Piotr Kołodziejczyk, Wojciech Kielan, Rossella Reddavid, Johanna W. van Sandick, Gian Luca Baiocchi, Ines Gockel, Andrew Davies, Bas P. L. Wijnhoven, Daniel Reim, Paulo Costa, William H. Allum, Guillaume Piessen, John V. Reynolds, Stefan P. Mönig, Paul M. Schneider, Elisenda Garsot, Emma Eizaguirre, Mònica Miró, Sandra Castro, Coro Miranda, Xavier Monzonis-Hernández, Manuel Pera

**Affiliations:** 1Hospital del Mar Research Institute (IMIM), Section of Gastrointestinal Surgery, Hospital del Mar, Department of Surgery, Universitat Autònoma de Barcelona, 08003 Barcelona, Spain; 1325220@uab.cat (M.D.C.);; 2Department of Pathology, Hospital Universitario del Mar, Cancer Research Program, Hospital del Mar Research Institute (IMIM), 08003 Barcelona, Spain; 3Section of Esophagogastric and Bariatric Surgery, Hospital Clinic, Department of Surgery, Universitat de Barcelona, 08193 Barcelona, Spain; 4Department of Surgery, General and Upper G.I. Surgery Division, University of Verona, 37126 Verona, Italy; 5Digestive Surgery, European Institute of Oncology, IRCCS, 20122 Milan, Italy; 6Department of GI Surgery, IRCCS, San Raffaele Hospital, Vita-Salute University, 20135 Milan, Italy; 7GB Morgagni-L Pierantoni Surgical Department, 47121 Forli, Italy; 8Department of Surgery, University Medical Center, 1007 Amsterdam, The Netherlands; 9Department of Surgical Oncology, Medical University of Lublin, 20-080 Lublin, Poland; 10Experimental Pathology and Therapeutics Group and Surgical Oncology Department, Portuguese Institute of Oncology, 4200-072 Porto, Portugal; 11Department of Surgery I, Jagiellonian University, 31-007 Krakow, Poland; 122nd Department of General and Oncological Surgery, Wroclaw Medical University, 50-367 Wroclaw, Poland; 13Department of Oncology, Division of Surgical Oncology and Digestive Surgery, University of Turin, San Luigi University Hospital, Orbassano, 10043 Turin, Italy; 14Department of Surgery, Netherlands Cancer Institute, Antoni van Leeuwenhoek Hospital, 1066 Amsterdam, The Netherlands; 15General Surgery Unit, Department of Clinical and Experimental Sciences, University of Brescia, ASST Cremona, 26100 Cremona, Italy; 16Department of Visceral, Transplant, Thoracic and Vascular Surgery, University Hospital of Leipzig, 04103 Leipzig, Germany; 17Department of Digestive Surgery, Guy’s & St Thomas’ National Health Service Foundation Trust, London SE1 7EH, UK; 18Department of Surgery, Erasmus University Medical Center, 3015 Rotterdam, The Netherlands; 19Department of Surgery, School of Medicine and Health, Technical University of Munich, 81675 Munich, Germany; 20Department of General Surgery, Faculdade de Medicina, Universidade de Lisboa, Hospital Garcia de Orta, 1649-028 Lisboa, Portugal; paulomatoscosta@gmail.com; 21Department of Surgery, Royal Marsden NHS Foundation Trust, London SW3 6JJ, UK; 22Department of Digestive and Oncological Surgery, University Lille, Claude Huriez University Hospital, 59037 Lille, France; 23Department of Surgery, Trinity College Dublin, St. James’s Hospital, D08 W9RT Dublin, Ireland; 24Division of Abdominal Surgery, University Hospital of Geneva, 1205 Geneva, Switzerland; 25Center for Visceral, Thoracic and Specialized Tumor Surgery, Hirslanden Medical Center, 5000 Zurich, Switzerland; 26Department of Surgery, Universitat Autònoma de Barcelona, Hospital Universitari Germans Trias i Pujol, 08916 Barcelona, Spain; 27Department of Surgery, Hospital Universitario de Donostia, 20014 Donostia, Spain; 28Department of Surgery, Hospital Universitari de Bellvitge, 08907 L’Hospitalet de Llobregat, Spain; 29Department of Surgery, Universitat Autónoma de Barcelona, Hospital Universitari Vall d’Hebron, 08035 Barcelona, Spain; 30Department of Surgery, Hospital Universitario de Navarra, 31008 Pamplona, Spain

**Keywords:** gastric cancer, gastrectomy, mortality, prediction, machine learning, validation

## Abstract

**Simple Summary:**

A 90-day mortality predictive model for curative gastric cancer resection based on the Spanish EURECCA Esophagogastric Cancer database was externally validated using the GASTRODATA registry. The externally validated model showed a modestly worse performance compared to the original model, nevertheless maintaining its discriminating ability in clinical practice.

**Abstract:**

Background: Radical gastrectomy remains the main treatment for gastric cancer, despite its high mortality. A clinical predictive model of 90-day mortality (90DM) risk after gastric cancer surgery based on the Spanish EURECCA registry database was developed using a matching learning algorithm. We performed an external validation of this model based on data from an international multicenter cohort of patients. Methods: A cohort of patients from the European GASTRODATA database was selected. Demographic, clinical, and treatment variables in the original and validation cohorts were compared. The performance of the model was evaluated using the area under the curve (AUC) for a random forest model. Results: The validation cohort included 2546 patients from 24 European hospitals. The advanced clinical T- and N-category, neoadjuvant therapy, open procedures, total gastrectomy rates, and mean volume of the centers were significantly higher in the validation cohort. The 90DM rate was also higher in the validation cohort (5.6%) vs. the original cohort (3.7%). The AUC in the validation model was 0.716. Conclusion: The externally validated model for predicting the 90DM risk in gastric cancer patients undergoing gastrectomy with curative intent continues to be as useful as the original model in clinical practice.

## 1. Introduction

Despite a significant decline in its incidence in recent years, gastric cancer remains the fourth leading cause of cancer death worldwide [1]. Surgical intervention continues to be the primary potentially curative option for patients with gastric cancer, even in the setting of multimodal treatment [2]. This intervention in benchmark patients is associated with an overall morbidity rate of 16.2% and with 30- and 90-day mortality rates of 0.3% and 0.5%, respectively [3]. Though, in other series, morbidity has risen to 20–45% [4,5,6,7] and mortality to 2–7% rates [4,6,7].

An accurate preoperative risk assessment for these procedures is important to help with the selection of patients. However, in gastric cancer surgery, few risk prediction models have been developed [8]. Most models focus on predicting survival following a curative resection, whereas only few studies have been conducted to predict operative mortality [9,10,11,12,13]. Moreover, the majority of these studies are based on classical logistic regression or Cox regression analysis, even though artificial intelligence (AI)-related tools are now available and being increasingly used to assist clinicians in providing tailor-made treatment decisions [14].

Additionally, it is important to mention that despite the growing number of predictive models (classical or developed with AI), their quality and clinical impact are often insufficient, also because of the lack of an external validation that would guarantee validity and clinical applicability [14]. The external validation of a risk prediction algorithm, in fact, is an important step in the process of building and evaluating a model, since it provides information about the reproducibility and generalizability of the model and assures its clinical applicability [14]. In gastric cancer surgery, only 13% of the predictive models developed have undergone a high-quality validation [8]. External validation is rarely performed because of its practical difficulty (need for multi-institutional collaboration across different geographic regions to achieve datasets of external cohorts in different settings) [15] and because of discriminative ability reduction in validation studies, which makes them unattractive for publication [8].

A clinical model for predicting the risk of 90-day mortality (90DM) after gastrectomy using AI was recently developed. The model showed an excellent performance (AUC 0.829) in the original cohort [16], but external validation of the risk prediction algorithm is necessary to provide information on its reproducibility and generalizability (or transportability), as well as to define its clinical applicability [14,17]. To our knowledge, external validation studies of ML models in the setting of gastric cancer surgery have not been previously reported [18].

The objective of the study was to perform an external validation of a 90DM risk prediction model using ML in gastric cancer patients undergoing gastrectomy with curative intent using a cohort from the European GASTRODATA database.

## 2. Materials and Methods

This study conformed to the TRIPOD10 (Transparent Reporting of a multivariable prediction model for Individual Prognosis or Diagnosis) reporting guidelines (Appendix A) [19].

### 2.1. Source of Data

#### 2.1.1. Study Development Cohort

The cohort for which the risk prediction model was derived has been previously described [16]. Briefly, data were retrieved from the Spanish EURECCA Esophagogastric Cancer Registry (SEEGCR) that covers data from 39 public hospitals of the National Health Care System from six regions in Spain, covering nearly a population of 14 million inhabitants. The SEEGCR database was audited for the 2014–2017 period with a completeness of 97% and data accuracy of 95% [20]. The SEEGCR is linked to the EURECCA Upper Gastrointestinal network, a multi-institutional population-based cohort registry that collects prospective clinical data from all patients with primary esophageal, gastro-esophageal junction (GEJ), and gastric cancer undergoing resection with curative intent.

#### 2.1.2. Validation Cohort

For the present study of multi-institutional validation, data were collected from the European GASTRODATA database. The registry collects retrospective and prospective clinical data from patients with primary gastric cancer, including cancer of the GEJ, that underwent surgical resection with curative intent between 2015 and 2022, in 25 hospitals from 11 European countries. As in the SEEGCR database, patients’ information was collected using an online platform (www.gastrodata.org, accessed on 5 September 2022) in which the following six sections had to be completed: (1) clinical features, (2) oncological characteristics and surgical data, (3) perioperative complications, (4) outcome at hospital discharge, and (5) outcome at 30 and 90 days postoperatively [5].

In fact, most variables used in the development of the model were also available in the GASTRODATA registry. Moreover, both registries used the same definition criteria for these variables, especially for those related to complications and outcome measures [21].

#### 2.1.3. Ethics

The local ethics committees of the centers participating in each of the registries (SEEGCR and GASTRODATA) approved the collection of anonymized data. The scientific committee of the GASTRODATA group approved sharing the dataset for the external validation project.

#### 2.1.4. Eligibility and Primary Outcome

All patients with primary gastric or GEJ cancer (excluding Siewert 1 tumors) who underwent gastrectomy (partial or total) with curative intent included in the GASTRODATA registry from 2015 to 2022 were eligible. The primary outcome was 90DM defined as all-cause mortality within 90 days after surgery.

#### 2.1.5. Predictor Characteristics and Statistical Analysis

The preoperative variables of the SEEGCR database used for the development of the original ML-based algorithm were also obtained from the GASTRODATA registry and compared each other. The principal investigators of the GASTRODATA centers were requested to retrieve some missing variables or variables not available in the registry, such as preoperative hemoglobin level and center volume. Age, body mass index (BMI), hemoglobin and albumin serum levels, and hospital volume activity (number of gastrectomies per center per year) were considered as continuous variables. The remaining variables (gender, BMI index, weight loss, ASA score, ECOG score, tumor location, clinical stage, neoadjuvant therapy, minimally invasive or open approach, subtotal or total gastrectomy, elective or urgent surgery, comorbidity as renal disease, pulmonary disease, peripheral vascular disease, myocardial infarction, diabetes mellitus, cerebrovascular disease, congestive heart failure, peptic ulcer disease, malignant lymphoma, dementia, liver disease, connective tissue disease, leukemia, hemiplegia, AIDS, malignant tumor, and metastatic tumor) were categorized as dichotomous variables by using one-hot encoding [22]. Missing data were imputed by including a separate category of predictor variables that had missing values [23]. Descriptive statistics are presented as means and standard deviations or numbers and percentages for continuous and categorical variables, respectively. Differences between the groups of patients who survived and those who died within 90 postoperative days were evaluated using the Fisher’s exact test for categorical variables or the Kolmogorov–Smirnov test for continuous variables. Statistical significance was set at *p* < 0.05.

#### 2.1.6. External Validation of the Predictive Model

Trained models developed in the previous study (Random Forest, cv-Enet, and glmboost, ensemble) [16] were used on the external validation set. Briefly, cv-Enet (Cross Validated Elastic net regularized logistic regression) [24] is an algorithm that determines the optimal coefficients for lasso and ridge penalties through internal cross-validation, whereas RF (Random Forest) and glmboost are composed of decision trees or a generalized linear model fitted with a boosting algorithm, respectively [25,26,27]. Finally, the ensemble model uses the 3 previous models combined with a linear blend of predicted probabilities using logistic regression. The discrimination of the models on the external validation dataset was assessed using the area under the curve (AUC). Sensitivity, specificity, positive predictive value (PPV), negative predictive value (NPV), and area under the precision–recall curve (AUPRC) were also reported for each model. In order to assess the feature attributions for each variable on the model testing, the “predict parts” function from the DALEX was used. [28]. For each sample, the absolute features’ attributions were calculated and averaged on the whole cohort. Data analysis was performed using R software version 4.2.0 (R Foundation for Statistical Computing, Vienna, Austria). The models were validated using mlr3 package [29].

The final model is freely available at https://gastrohmar.shinyapps.io/rf_eurecca_model/ (accessed on 1 July 2024).

## 3. Results

A total of 2595 patients from 25 hospitals in 11 European countries were included in the GASTRODATA database over an 8-year period (2015–2022), with 90-day follow-up available for all patients. Patients from the Hospital del Mar registered in the GASTRODATA registry were excluded because they were part of the development cohort. Finally, 2546 patients from 24 hospitals in 11 European countries were included for the analysis (Appendix A). The overall rate of missing data for variables was 4% (3215 items in 86,564 cells). The most frequently missing characteristics were preoperative albumin (*n* = 668 [26%]) and Eastern Cooperative Oncology Group (ECOG) score performance status (*n* = 554 [21%]).

Table 1 shows data on the preoperative variables for the development and external validation cohorts. The mortality rate in the GASTRODATA cohort was lower than that in the SEEGCR, indeed, 3.7% (95 patients) versus 5.6% (179 patients) of the SEEGCR died within 90 days. Age, BMI, and the rates of congestive heart failure, chronic obstructive pulmonary disease (COPD), cerebrovascular disease, complicated diabetes mellitus, leukemia, malignant lymphoma, and liver disease were significantly lower in the GASTRODATA cohort. Furthermore, the GASTRODATA patients more frequently had a lower ECOG performance status and American Society of Anesthesiologists (ASA) score, with higher percentages of weight loss and more advanced clinical T and N stages. Regarding the localization of the tumor, there were more cases of linitis plastica and GEJ tumors. Additionally, elective and open procedures were more commonly performed in the external validation cohort, as well as neoadjuvant treatment and total gastrectomy. The mean volume of the centers was higher in the external validation cohort.

### 3.1. Model Performance: Discrimination

Table 2 summarizes all the precision metrics obtained with the random forest model, which was the model with the best performance both on the development and the external validation cohorts (Figure 1). The AUCs for the development and external validation cohorts were 0.844 and 0.716, respectively, leading to a 11.3% performance reduction. The precision metrics obtained with the other models (cv-Enet, glmboost, and ensemble) are shown in Appendix A.

### 3.2. Variable Importance

A feature attribution analysis on the external validation dataset was assessed by decomposing the model predictions using variable-attribution measures that could be assigned to specific variables. The most important factors for the prediction were age, ASA score, volume center, preoperative serum albumin level, ECOG, preoperative serum hemoglobin level, and neoadjuvant treatment (Figure 2).

## 4. Discussion

We conducted an external validation of the ML-based SEEGCR risk prediction model of 90DM on patients undergoing gastric cancer resection with curative intent using the GASTRODATA registry, a large multicenter European database. To our knowledge, this is the first external validation study of an ML-based model for the prediction of mortality in the field of gastric cancer surgery. The AUC for the external validation cohort was 0.716, which is lower than those achieved previously on the development (0.844) and internal–external validation (0.829) cohorts. However, this drop in performance may not invalidate the usefulness of having available an additional tool for assessing the prognosis of surgical patients with gastric cancer.

The external validation of a risk prediction algorithm is important to assess the clinical applicability of the model in similar (reproducibility) or different populations (generalizability or transportability) [14,17]. Despite the growing interest in developing predictive models in clinical practice, a recent review provided a summary of the state of the art of AI-enabled decision support in surgery and found that, among 36 studies, external validation was performed in only 5 of them (13.8%) [18]. In the field of esophagogastric cancer surgery, the discriminative ability of models was significantly lower in the validation than in the development phase [8]. In an evaluation of the external validation processes of 31 prediction models of different conditions (cardiovascular diseases, gastrointestinal-related diseases, malignancies, and other) [30], it was shown that the AUC decreased on average by 0.062, which, in fact, would be quite similar to the AUC higher than 0.716 found in our study. The limited number of external validation studies may be explained by two reasons, such as difficulties in obtaining external cohorts with a sufficiently large sample size and the performance of the validation model with a discriminating ability usually being inferior to that found in the development model.

A collaboration between the SEEGCR and the GASTRODATA registry allowed us to use their dataset with 2546 cases for the external validation, which conforms to the recommendation of having a cohort of at least 1000 patients for the validation [18]. However, both registries present differences. First, the SEEGCR is a population-based registry that includes all consecutive patients operated on in all centers from six Spanish regions, representing real-world practice, whereas the GASTRODATA registry includes a selection of patients operated on in 25 medium- and high-volume hospitals from 11 European countries. Second, an overall assessment of the relatedness between the development and the external validation samples revealed case mix differences of predictor variables, as well as a different outcome (90DM) occurrence. While patients in the SEEGCR appeared to be in poorer physical conditions (older, worse ECOG and ASA scores, and more comorbidities), patients in the GASTRODATA cohort had more advanced clinical T and N stages, more frequently received neoadjuvant treatment, and had more elective and open procedures, with total gastrectomy as the most common procedure. Additionally, the mean volume of the participating hospitals was significantly higher in the external validation cohort. The mortality rates in the GASTRODATA registry were lower than those in the SEEGCR registry. This may be explained by the higher volume of hospitals contributing to the GASTRODATA as compared to the heterogeneity of the volume and technologic level of hospitals participating in the SEEGCR [20].

It is still important to note that the AUC alone may not provide a complete picture of the predictive performance of a model, as it does not take into account factors such as the model calibration or prevalence of the outcome being predicted. Therefore, it is typically recommended to consider other performance metrics in addition to the AUC, such as sensitivity, specificity, predictive values, and calibration measures [18]. Another important performance metric is the area under the precision–recall curve (AUPRC), which is based on the PPV value and sensitivity and evaluates how well a model can identify positive examples in a dataset. The importance of AUPRC relies on the fact that it maintains its strength even under imbalanced datasets, mostly in datasets in which relatively rare events are predicted [31,32]. Based on metrics data, the RF model is the best model to identify patients at risk of 90DM, as it showed the highest PPV together with the lowest sensitivity and the highest AUPRC in the GASTRODATA cohort.

The current study provides insights into the additional value of particular input variables to predict the risk of 90DM. The differences between the values of the variables detected in the development and validation cohorts were minimal, and four of the most important factors (age, volume, and preoperative serum levels of hemoglobin and albumin) were shared by the two cohorts. These four variables were also clinically relevant and easy to obtain at the bedside.

Several potential limitations of the study are noted. First, the GASTRODATA registry includes a selection of patients undergoing gastrectomy at the different participating hospitals, and not all patients were consecutively recruited (it has been estimated that 396 cases are missing based on the mean real volumes reported by each hospital). Secondly, there was a difference in the quality of the datasets. Indeed, the GASTRODATA has not undergone an audited process, in contrast to the SEEGCR registry that was audited (period 2014–2017) with a 97% and 95% of completeness and data accuracy, respectively [20]. A third limitation is the overall rate of missing data of 4% in the GASTRODATA dataset (3215 items in 86,564 cells) and 0.6% (677 items in 101,824 cells) in the SEEGCR. This higher rate of missing data could also be explained due to some differences in the classification of variables. For example, in GASTRODATA, the variables “leukemia” and “malignant lymphoma” were collected as the same variable, and the option “cNx” in “Tumor cN stage” was not considered in SEEGCR. In both cases, data were recorded as missing. Additionally, it should be noted that 11.8% of the validation cohort were classified as ASA I. It is probable that ASA scores would have been underestimated because patients with cancer may fit in the ASA II score as they already have a systemic disease. A fourth limitation is the few events in the external validation cohort, 95 deaths at 90 days (compared with 179 of the SEEGCR), at the threshold of the minimum required number of events (100) and well below the optimal number (>250) [14].

## 5. Conclusions

In conclusion, the ML-based algorithm of the SEEGCR registry for predicting the risk of 90DM in patients undergoing gastric cancer surgery with curative intent performed modestly worse in a European multi-institutional-based external validation study. However, the predictive model continues to be useful to assess the post-surgical clinical outcome in this population. The external validation of the 90DM predictive model adds value to the original instrument.

## Figures and Tables

**Figure 1 cancers-16-02463-f001:**
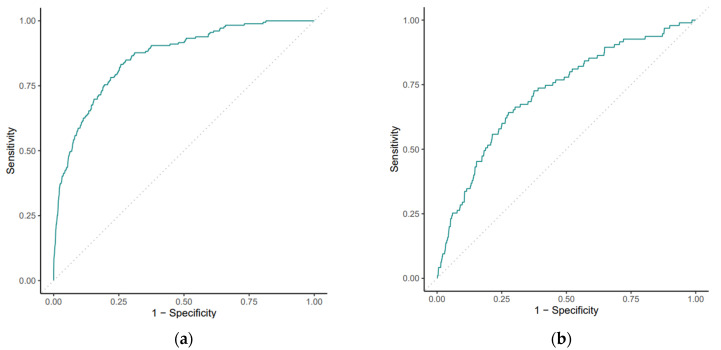
Model discriminations in both the development (**a**) and the external validation (**b**) cohorts. The AUC for random forest (RF) model in the development cohort was 0.844 (95% confidence interval [CI] 0.84–0.85) as compared with an AUC of 0.716 (95% confidence interval [CI] 0.66–0.77) of the external validation cohort.

**Figure 2 cancers-16-02463-f002:**
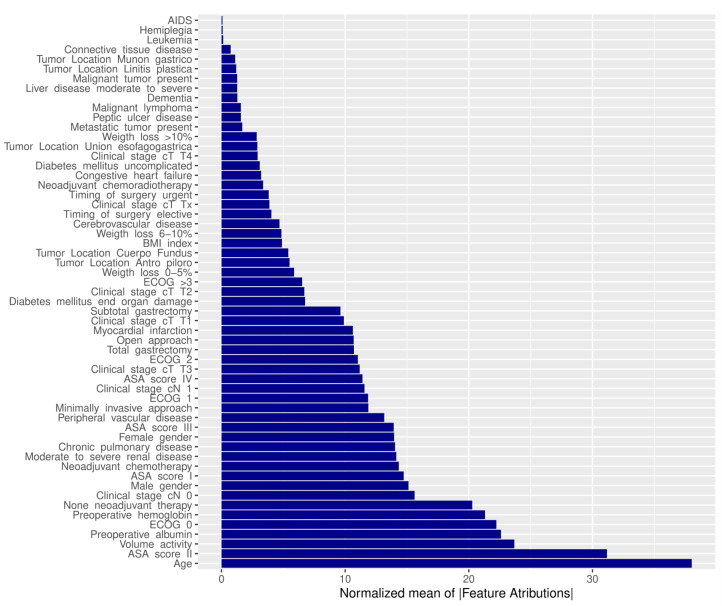
Feature attribution of RF model. Normalized mean of absolute feature attributions of all factors of the GASTRODATA cohort on the random forest (RF) model.

**Table 1 cancers-16-02463-t001:** Potential risk factors for 90-day mortality in the development and external validation cohorts.

	Development Cohort (*n* = 3182)	Validation Cohort (*n* = 2546)	*p* Value
Sex, *n* (%)			0.252
Male	1978 (62.1)	1544 (60.6)	
Female	1204 (37.9)	1002 (39.4)	
Age, years, mean (SD)			
Body mass index ^$^, kg/m^2^, mean (SD)	26 (4.6)	25 (4.5)	<0.001
Missing	90 (2.8)	94 (3.7)	
ECOG performance status, *n* (%)			<0.001
0	1185 (37.2)	1203 (47.2)	
1	1661 (52.2)	626 (24.6)	
2	269 (8.5)	118 (4.6)	
>3	51 (1.6)	45 (1.8)	
Missing	16 (0.5)	554 (21.8)	
ASA index, *n* (%)			<0.001
I	110 (3.5)	300 (11.8)	
II	1435 (45.0)	1288 (50.5)	
III	1510 (47.5)	878 (34.5)	
IV	127 (4.0)	55 (2.2)	
Missing, *n* (%)	0 (0)	25 (1.0)	
Weight loss ^$^, %, *n* (%)			<0.001
0–5%	2164 (68.0)	1368 (53.7)	
6–10	603 (19.0)	646 (25.4)	
>10%	390 (12.3)	370 (14.5)	
Missing	25 (0.7)	162 (6.4)	
Preoperative hemoglobin level, g/dL, mean (SD)	12.0 (1.9)	12.0 (2.1)	<0.038
Missing, *n* (%)	24 (0.8)	470 (18.5)	
Preoperative albumin level, mg/dL, mean (SD)	38 (6.2)	38 (6.4)	0.431
Missing, *n* (%)	441 (13.9)	668 (26.2)	
Myocardial infarction, *n* (%)			0.653
Yes	253 (8.0)	193 (7.6)	
No	2929 (82.0)	2348 (92.2)	
Missing	0 (0)	5 (0.2)	
Congestive heart failure, *n* (%)			0.026
Yes	183 (5.8)	112 (4.4)	
No	2999 (94.2)	2429 (94.4)	
Missing	0 (0)	5 (0.2)	
Chronic pulmonary disease, *n* (%)			<0.001
Yes	450 (14.1)	246 (9.7)	
No	2732 (85.9)	2295 (90.1)	
Missing	0 (0)	5 (0.2)	
Connective tissue disease, *n* (%)			0.647
Yes	47 (1.5)	33 (1.3)	
No	3135 (98.5)	2508 (98.5)	
Missing	0 (0)	5 (0.2)	
Peripheral vascular disease, *n* (%)			0.098
Yes	226 (7.1)	211 (8.3)	
No	2956 (92.9)	2330 (91.5)	
Missing	0 (0)	5 (0.2)	
Cerebrovascular disease, *n* (%)			0.021
Yes	200 (6.3)	123 (4.8)	
No	2982 (93.7)	2420 (95.1)	
Missing	0 (0)	3 (0.1)	
Dementia, *n* (%)			0.944
Yes	33 (1.0)	25 (1.0)	
No	3149 (99.0)	2518 (98.1)	
Missing	0 (0)	3 (0.1)	
Peptic ulcer disease, *n* (%)			0.214
Yes	158 (4.9)	146 (5.7)	
No	3024 (95.1)	2397 (94.2)	
Missing	0 (0)	3 (0.1)	
Diabetes mellitus (uncomplicated), *n* (%)			1.000
Yes	519 (16.3)	414 (16.3)	
No	2663 (83.7)	2127 (83.5)	
Missing	0 (0)	5 (0.2)	
Diabetes mellitus (end-organ damage), *n* (%)			<0.001
Yes	137 (4.3)	39 (1.5)	
No	3045 (95.7)	2499 (86.4)	
Missing	0 (0)	0 (0.3)	
Leukemia, *n* (%)			0.002
Yes	16 (5.0)	0 (0)	
No	3166 (99.5)	2193 (86.1)	
Missing	0 (0)	353 (13.9)	
Malignant lymphoma, *n* (%)			<0.001
Yes	34 (1.1)	0 (0)	
No	3148 (98.9)	2193 (86.1)	
Missing	0 (0)	353 (13.9)	
Liver disease/moderate to severe, *n* (%)			<0.001
Yes	82 (2.6)	0 (0)	
No	3100 (97.4)	2526 (99.2)	
Missing	0 (0)	20 (0.8)	
Hemiplegia, *n* (%)			1.000
Yes	8 (0.3)	6 (0.2)	
No	3174 (99.7)	2537 (99.6)	
Missing	0 (0)	3 (0.2)	
Metastatic tumor present, *n* (%)			1.000
Yes	36 (1.1)	28 (1.1)	
No	3146 (98.9)	2513 (98.7)	
Missing	0 (0)	5 (0.2)	
Moderate to severe renal disease, *n* (%)			0.654
Yes	162 (5.1)	137 (5.4)	
No	3020 (94.9)	2404 (94.4)	
Missing	0 (0)	5 (0.2)	
AIDS, *n* (%)			0.453
Yes	6 (0.2)	2 (0.1)	
No	3176 (99.8)	2539 (99.7)	
Missing	0 (0)	5 (0.2)	
Timing of surgery, *n* (%)			<0.001
Elective	3002 (94.3)	2476 (97.2)	
Emergency	180 (5.7)	68 (2.7)	
Missing	0 (0)	2 (0.1)	
Tumor location, *n* (%)			<0.001
Antrum-pylorus	1276 (48.1)	1212 (47.6)	
Corpus-fundus	76 (40.1)	848 (33.3)	
Linitis plastica	33 (1.0)	86 (3.4)	
Stump	81 (2.6)	0 (0)	
Gastro-esophageal junction	259 (8.1)	348 (13.7)	
Missing	3 (0.1)	52 (2.0)	
Tumor cT stage ^&^, *n* (%)			<0.001
T1	528 (16.6)	235 (9.2)	
T2	792 (24.9)	447 (17.6)	
T3	1082 (34.0)	1095 (43.0)	
T4	569 (17.9)	544 (21.4)	
Tx	173 (5.4)	206 (8.1)	
Missing	38 (1.2)	19 (0.7)	
Tumor cN stage ^&^, *n* (%)			<0.001
Negative	1771 (55.7)	858 (33.7)	
Positive	1377 (43.3)	1299 (51.0)	
Missing	34 (1.0)	389 (15.3)	
Neoadjuvant therapy, *n* (%)			<0.001
None	2232 (70.1)	1383 (54.3)	
Chemoradiotherapy	54 (1.8)	46 (1.8)	
Chemotherapy	888 (27.9)	1117 (43.9)	
Missing	8 (0.2)	0 (0)	
Surgical approach, *n* (%)			<0.001
Open	1706 (53.6)	1884 (74.0)	
Laparoscopic	1476 (46.4)	662 (26.0)	
Type of gastrectomy, *n* (%)			<0.001
Partial	1818 (57.1)	1211 (47.6)	
Total	1364 (42.9)	1331 (52.3)	
Missing	0 (0)	4 (0.1)	
Volume activity, mean/year/hospital, mean (SD)	24 (10)	60 (49)	<0.001
90-day mortality, *n* (%)	179 (5.6)	95 (3.7)	<0.001

^$^ At the time of diagnosis; ^&^ According to the seventh edition of the AJCC; AIDS indicates acquired immune deficiency syndrome; ASA, American Society of Anesthesiologists; ECOG, Eastern Cooperative Oncology Group; SD, standard deviation.

**Table 2 cancers-16-02463-t002:** Performance metrics from the development and external validation cohorts for the Random Forest (RF) model.

Metrics	Development Cohort	External Validation Cohort
AUC	0.829 (95% CI 0.743–0.916)	0.716 (95% CI 0.663–0.769)
Sensitivity	0.125 (95% CI 0.016–0.383)	0.074 (95% CI 0.030–0.146)
Specificity	0.979 (95% CI 0.953–0.993)	0.984 (95% CI 0.979–0.989)
PPV	0.286 (95% CI 0.037–0.710)	0.156 (95% CI 0.065–0.295)
NPV	0.945 (95% CI 0.909–0.969)	0.965 (95% CI 0.957–0.972)
AUPRC	0.253	0.093

Abbreviations: AUC, Area under the curve; PPV, positive predictive value; NPV, negative predictive value; AUPRC, area under the precision recall curve; and CI: confidence interval.

## Data Availability

All code and scripts to reproduce the experiments of this paper are available at: https://github.com/Tato14/rf_gastro accessed on 1 July 2024.

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
