# Peer review of "International External Validation of Risk Prediction Model of 90-Day Mortality after Gastrectomy for Cancer Using Machine Learning"

_cancers, 2024, doi:10.3390/cancers16132463_

Round 1

Reviewer 1 Report

Comments and Suggestions for Authors

This manuscript demonstrates that the validated model can predict the risk of gastric cancer survival. Although the accuracy is somewhat poor, the AUC of 0.716 is still considered to be acceptable. The methodology is excellent and the number of cases is sufficient. The discussion is very detailed and complete. Therefore, I think this manuscript should be accepted.

Comments on the Quality of English Language

I think that minor editing of English required in this manuscript.

Reviewer 2 Report

Comments and Suggestions for Authors

Congratulations to Dal Cero et al. on their interesting article. The authors conducted an external validation of a prediction model for 90-day mortality following gastrectomy for cancer. This study is significant and has the potential to receive numerous citations, contributing substantially to the journal's impact factor.

I have some remarks:

Title: No comments.

Abstract: The authors should enhance the abstract, as it serves as the gateway that determines whether readers will engage with the article. This is particularly true for the conclusion. The current conclusion does not seem appealing and appears to undervalue the study's importance.

Introduction: The introduction is too brief. I suggest expanding the justification for the study to provide a clearer context and significance.

Methods: I recommend a more detailed description of the predictive model rather than just citing the original study. It would be helpful to detail the variables included in the model, as readers may not have access to the original study for reference.

Results: Please edit Figure 2 to improve readability. The use of underlines after every word is unnecessary and distracting.

Discussion: A specific limitation of the study should be addressed. The ASA score is a critical component of the predictive model in this cancer study. However, there is limited information on how the ASA was defined across different institutions. Notably, 11.8% of the validation cohort was classified as ASA I. A common mistake in clinical practice is to score the ASA “as if the patient did not have cancer,” despite cancer being a systemic disease. Weight loss, anemia, previous chemotherapy, or decreased performance status should warrant a higher ASA score. There may have been an underestimation of ASA scores, particularly in the validation cohort.

Conclusion: The current conclusion does not adequately reflect the value of the study. It is expected that performance will decrease in the validation cohort compared to the internal validation. Additionally, I suggest moving the link to the predictive model from the conclusion to the methods section.
